# Synthetic Haemostatic Sealants: Effectiveness, Safety, and In Vivo Applications

**DOI:** 10.3390/ph17030288

**Published:** 2024-02-23

**Authors:** Federica Curcio, Paolo Perri, Paolo Piro, Stefania Galassi, Roberta Sole, Sonia Trombino, Roberta Cassano

**Affiliations:** 1Department of Pharmacy, Health and Nutritional Science, University of Calabria, Arcavacata, 87036 Rende, Italy; federica.curcio@unical.it (F.C.); roberta.sole@unical.it (R.S.); 2Complex Operating Unit Vascular and Endovascular Surgery, Annunziata Hospital, 1 Via Migliori, 87100 Cosenza, Italy; p.perri@aocs.it (P.P.); p.piro@aocs.it (P.P.); 3Complex Operating Unit Interventional Radiology, Annunziata Hospital, 1 Via Migliori, 87100 Cosenza, Italy; s.galassi@aocs.it

**Keywords:** tissue adhesive, cyanoacrylate-based glues, polysaccharide hydrogel-microsphere powder, vascular application

## Abstract

Rapid haemostasis during surgery is essential when one wants to reduce the duration of operations, reduce the need for transfusions, and above all when one wants to achieve better patient management. The use of haemostatic agents, sealants, and adhesives improves the haemostatic process by offering several advantages, especially in vascular surgery. These agents vary widely in their mechanism of action, composition, ease of application, adhesion to wet or dry tissue, immunogenicity, and cost. The most used are cyanoacrylate-based glues (Glubran 2) or polysaccharide hydrogel-microsphere powder (Arista^TM^AH). This work is based on a retrospective study carried out on a sample of patients with different vascular diseases (FAV, pseudoaneurysm, and PICC application) in which two different haemostatic sealants were used. The aim was to assess the safety, the advantages, and the ability of both sealants to activate the haemostatic process at the affected site, also in relation to their chemical-physical characteristics. The obtained results showed that the application of Glubran 2 and Arista^TM^AH as surgical wound closure systems is effective and safe, as the success achieved was ≥94% on anastomoses of FAV, 100% on stabilization of PICC catheters, and ≤95% on pseudoaneurysms.

## 1. Introduction

The tissue adhesives are biomaterials used in the treatment of haemostasis, wound closure, and tissue repair [1]. Compared to sutures, the advantages of using these adhesives relate to their ability to polymerize in situ, to adapt to complex wound contours and dimensions (sufficient mechanical flexibility), to be biocompatible, to possess adequate mechanical and physical properties (such as burst strength, tensile strength, and shear strength and shear resistance), to have high bond strength to moist tissue or organs, and finally to be easily applied [2,3]. They fall into two categories: (i) tissue adhesives based on natural polymers (fibrin, albumin, and gelatin); and (ii) synthetic tissue glues, based on cyanoacrylate, poly(ethylene glycol) (PEG), catechol, and methacrylic anhydride [4,5,6]. The design of an effective tissue adhesive is related to a careful tissue–adhesive interface that includes an adhesion layer, where the adhesive makes intimate contact with the tissue, and an adhesive matrix, which consists of the polymer network of the substances that make up the adhesive. The adhesion layer anchors to the underlying tissue through chemical, physical, or covalent and non-covalent bonds, while the adhesive matrix provides structural support to the adhesion layer and determines the chemical and physical characteristics of the adhesive [7]. From the 1950s onwards, cyanoacrylate glue was only used for wound closure; in 2000 it was finally approved for internal use in the treatment of arteriovenous malformations as a liquid gel embolic system [6]. Cyanoacrylate comes in four forms: methyl-2-cyanoacrylate (2-MCA), ethyl-2-cyanoacrylate (2-ECA), n-butyl-2-cyanoacrylate (nBCA), isobutyl-cyanoacrylate (ICA), and 2-octyl-cyanoacrylate (2-OCA). The short-chain forms (2-MCA, 2-ECA) are rarely used due to their rapid degradation and toxic effects, whereas the long-chain forms (nBCA, 2-OCA) are the most widely used in industry and medicine because they provide a strong and rigid bond in contact with tissues [8]. In fact, having a consistency like that of a liquid gel at room temperature—when it comes into contact with basic substances (such as water, blood, body tissue or moisture) or negatively charged ions—it undergoes an exothermic polymerization that hardens it into a solid adhesive film [9,10]. The cyanoacrylate glue used in this work is Glubran 2, a CE-certified Class III surgical medical device for internal and endovascular use [11,12]. This device has undergone preclinical and clinical testing to assess its safety. In particular, it has undergone biocompatibility testing, performance testing, and animal studies (depending on the application). Biocompatibility tests are trained according to Good Laboratory Practice (GLP; 21 CFR 58) in accordance with International Organization for Standardization (ISO) 10993 [13]. Laboratory tests demonstrating safety and efficacy prior to animal studies include the following: cytotoxicity—International Organization for Standardization (ISO10993-5), sensitization and irritation (ISO 10993-10), implantation (ISO 10993-6), pyrogenicity, acute, subchronic, and chronic toxicity (ISO10993-11), haemolysis (ISO 10993-4), and genotoxicity (ISO 10993-3). Animal studies for specific applications of Glubran have been conducted on sheep, dogs, rats, mice, and pigs. In addition to these tests, as bioadhesives in contact with blood degrade with time, degradation studies and blood compatibility studies have been performed on them [14]. In particular, in the case of Glubran 2, several models were developed to mimic different clinical situations; for example, the ascending pharyngeal artery in the pig serves as a model for arteriovenous malformations (AVMs). Aneurysms, including aortic aneurysms, have been created in rabbits and pigs and arteriovenous fistulas have been created in dogs [15].

This device has a high adhesive capacity in a humid environment (viable tissue) with haemostatic, sealing, and bacteriostatic (barrier) properties, and high elasticity and tensile strength for waterproof and breathable bonding, as well as a low polymerization temperature (45 °C) that starts on contact with the tissue (1–2 s) and is complete in 60–90 s [16,17]. It also possesses high biocompatibility and slow biodegradability, does not cause the release of toxic products, and the minimum use quantity is 1 mL T20 cm^2^. The film formed after the application of this glue can be easily punctured by suture needles, and the glue when mixed with Lipiodol^®^ can be opacified and its rate of polymerization can be altered, favouring the formation of a less uniform and more flocculent polymer with a gel-like consistency (embolizing agent) [18]. Glubran 2 is commercially available as ready-to-use disposable applicator devices, formulated as a clear liquid in single-dose bottles of 0.25, 0.50, and 1 mL to be stored between 2 and 8 °C. Blood vessels that are subjected to Glubran 2 injection are embolized via three mechanisms: (1) cast and thrombus formation [19,20], (2) the adhesion to the inner vascular wall [21], and (3) damage to the vascular endothelium caused by an inflammatory response [22]. Another device used in the vascular field with haemostatic action is the Arista^TM^AH, an absorbable topical haemostatic that uses hydrophilic polysaccharide haemosphere (MPH) technology to help control blood loss [23]. The product is derived from purified vegetable starch and is supplied as a powder substance with a variety of applicators. Starch is an abundant hydrophilic natural biopolymer composed of anhydroglucose units that have hydrophilicity, biodegradability, biocompatibility, and similarity to the skin extracellular matrix, making them useful for various biomedical applications. Hydrophilicity and biodegradability are two crucial properties of starch granules that belong to the passive haemostats class and act as a “molecular sieve” by extracting fluids and blood [24,25]. Instead, the Arista^TM^AH haemostatic activity is activated when, upon meeting the liquid components of the blood, they are absorbed by the polysaccharide hydrogel-haemospheres and are concentrated to form a molecular network (gel) that allows efficient thrombus formation through natural coagulation (Figure 1). In fact, in situ gelling polymeric systems having functional groups that can react with functional groups present in tissue, have been found to be adhesive in nature [26]. This device has been approved by the FDA precisely because it enhances the biological process of coagulation through the formation of a natural haemostatic plug. Also, in the case of Arista^TM^AH, clinical and preclinical tests to assess its safety are based on biocompatibility tests, performance tests, and animal studies. Animal studies are mainly carried out on 6–8 week old Sprague–Dawley rat models, in which the surgical site is performed on the abdominal area. Tests showed no signs of inflammation at any time after application of the powder [27]. The aim of this work was to evaluate the effectiveness and safety of the two devices in the surgical field by differentiating their application based on the pathology treated and the haemostasis mechanism exerted on it.

## 2. Results

### 2.1. Study Population

During the interval (January 2023–September 2023), 60 patients (*n* = 28 picc, *n* = 20 with radial or femoral pseudoaneurysms, and *n* = 33 for arteriovenous fistula packing for haemodialysis) were subjected to Glubran 2 and Arista^TM^ AH haemostatic agent application. The mean age of the patients was about 70 years, the mean INR was 1.3, and 56% of the patients were on antiplatelet or anticoagulant therapy (cardioaspirin + xarelt, cardirene, duoplavin), 20% on anticoagulant therapy (low molecular weight heparin or New Oral Anticoagulants or NAO), and 24% on single antiplatelet therapy (Table 1).

### 2.2. Procedure Data

The most used surgical access for haemodialysis fistula placement is the cephalic radius, to be performed (for the same vein calibre) on the arm least used by the patient (if right-handed, left-handed) (DX 70% of patients, SX 30% of patients). The major site of pseudoaneurysm is the left radial artery (site of interventional cardiologist access) and right femoral artery (site of endovascular access for major endovascular procedures—EVAR aneurysms or carotid artery CAS). In most cases of pseudoaneurysm formation, the most frequently used introducer was the 6 Fr (68% of cases) on the common femoral artery and proximal radial artery, 7 Fr on the common femoral artery and distal radial artery (22% of cases), and 10 Fr (10% of cases) on the common femoral artery. For patients to undergo arteriovenous fistula in the operating room, an antihypertensive (catapresan used) was administered to maintain adequate pressures for opening the artery and packing the fistula, sodium heparin to avoid clots during clamping of the artery, and a pre-operative antibiotic as per protocol to avoid infection. For patients undergoing pseudoaneurysm exclusion, a pre-operative antibiotic and antihypertensive were administered for the same reason as before. Both patients were monitored with continuous electrocardiogram (ECG) and continuous saturation through devices in the anaesthetist’s charge. For patients with peripherally inserted central catheters (PICC), the procedure does not require a systemic antibiotic, instead it requires ECG monitoring to recognize *P*-wave variation during catheter placement, a sealed gown for the angio-suite procedure with Rx exposure, or operator sterility if only ultrasound guided. All three procedures require local anaesthetic (Lidocaine). The choice of sealant for the closure of a lesion was site specific; therefore, in this study, Glubran 2 was chosen to ensure haemostasis on arterial lesions with high-pressure blood flow (~120 mmHg) typical of a declamped artery and to prevent the occurrence of infection related to the presence of a PICC between different layers of skin. Gluing carried out exclusively with glue must effectively resist the physiological loads that tend to pull the tissues away from each other, thus ensuring an even distribution of loads in all affected areas, without compromising the elastic properties of natural fabrics. Glubran 2 glue was found to have a broad spectrum activity against Gram-positive and Gram-negative bacteria, yeast, and fungi. The deformation and mechanical pressure exerted by the vascular tissue emphasize the importance of the adhesive’s fatigue strength. The choice to use Arista^TM^AH was made, considering the ability of this sealant to act on larger tissue lesions with tassel bleeding, that is, when profuse bleeding occurs at low pressure on an iatrogenic lesion of a parenchymatous tissue. In addition, it is completely degraded within 24–48 h in vivo.

### 2.3. Outcomes and Follow-up

In the use of Glubran 2, the primary success resulting from the application of this sealant was ≥94% on anastomoses of arteriovenous fistulas, equal to 100% on the stabilization of catheters from PICC, and ≤95% on pseudoaneurysms. Failures, on the other hand, were related to active bleeding after 3 min or secondary bleeding in bedside drainage as a result of heparin overload (≥5000 I.U.) or the use of antiplatelet/anticoagulants (not discontinued by the patient before the procedure). In the use of Arista^TM^AH, the success resulting from its use was ≥96% on anastomoses of arteriovenous fistulas, demonstrating that, unlike glue, the powder’s large specific surface area, high porosity, and exceptional water absorption rate are more effective.

Specifically, suction drainage was applied to all patients undergoing FAV and pseudoaneurysm surgery for continuous monitoring of the surgical site and any blood loss. This drain has been used in dialysis and non-dialysis patients, in treatment with or without NAO, and/or EBPM via os administration, in treatment with or without antiaggregant therapy via os administration. The device is equipped with a lateral centimetric guide for the exact quantification of any blood loss. The nurses note at 1, 3, 6, and 24 h the amount present as shown in Table 2 and Table 3, which, if less than 10 cc, is indicative of no bleeding. The results showed that 96% of patients treated with Arista^TM^AH after FAV surgery were discharged after 24 h due to the absence of bleeding from the surgical site (Table 2a,b); 94% of patients treated with Glubran 2 after FAV surgery were discharged after 24 h due to the absence of bleeding (Table 2a,b). Other outcome measures in FAV patients treated with Arista^TM^AH or Glubran 2 were the presence of haematoma at the surgical site and the wound closure process. The results in Table 2a show that 50% of patients haemodialysed at 24 h had haematoma at the surgical site in both sealant treatments, especially when subjected to therapy with anti-aggregant or NAO/EBPM; only 10% had it at 7 days. Healing with follow-up of the surgical wound at 7 days occurred in 95% of the cases. No patients experienced any adverse effects in either procedure.

In the case of patients undergoing pseudoaneurysm treatment, the success rate obtained is relative to the use of Glubran 2 alone. Again, patients on dialysis and off-dialysis, treated with or without NAO and/or EBPM via os administration, or treated with or without antiaggregant, were taken into account. After emptying the pseudoaneurysm, a 6-mm PVC drain was applied under suction to monitor postoperative blood loss (using an Ulmer PVC drainage catheter without needle guide). The results shown in Table 3a,b revealed that 93% of the patients treated with Glubran 2 were discharged after 24 h due to the absence of bleeding from the surgical site (≤10 cc), all patients had a haematoma at the surgical site (of which 50% maintained it at 7 days (antiplatelet and NAO/EBPM therapy patients)), while at the 30-day follow-up it had disappeared in all patients. The type of suture used (Vycril or Metal clips) was also taken into account in the follow-up of wound closure and stitch removal. In 95% of the patients in whom a vycril suture was applied, healing occurred at 7 days after the operation, only in 5% did no healing occur (anti-clotting and NAO/EBPM therapy patients). In 90% of patients with metal clips, healing occurred 7 days after surgery, with only 10% not healing (anti-clotting and NAO/EBPM therapy patients).

Neither adhesive induced thrombosis within the arterial or venous vessels with which they came into contact at the treatment site (artery or vein for FAV or vein for PICC apposition, or artery as in the case of the pseudoaneurysm). This result emerged after an intra-operative postoperative echocolour doppler was performed with a Linear L7-11 MHz probe covered by a sterile protective device applied on it.

### 2.4. Haemostasis Evaluation

At the end of the 10 min observation period, the haemostatic success of lesions treated with Arista^TM^AH was 94%. In most cases of Arista^TM^AH failure, blood leaked from under a contiguous gel mass or occurred in patients with minimal INR alteration or the use of antiplatelets (not discontinued prior to surgery) or anticoagulants. Controlling for differences in the level of pre-treatment bleeding, the median time to haemostasis for Arista^TM^AH was approximately 5 min after its use combined with gentle manual compression by the surgeon. At the end of the 10 min observation period, the haemostatic success of lesions treated with Glubran 2 was 98%. In the majority of cases of Glubran 2 failure, blood leaked between two or more Prolene stitches at the suture site in patients with non-discontinued anticoagulant therapy (NAO) in favour of low molecular weight heparin (EBPM). Controlling for differences in pre-treatment bleeding levels, the median time to haemostasis for Glubran 2 was about 2 min after its use with a 5 min observation time (*p* ≤ 0.05) (Table 4).

## 3. Discussion

An ideal sealant, in order to guarantee effective haemostasis in all types of patients undergoing vascular procedures, should be used individually, be atraumatic without foreign bodies or mechanisms that alter the site it acts on, but above all it should be easy to use, have a success rate of ≥95%, and be cost-effective. In this retrospective study, we have focused our attention on the application and the efficacy of two sealants in three different typologies of vascular surgery and in a specific category of patients (Figure 1). The obtained results show us that the two haemostatics Glubran 2 and Arista^TM^AH prove to be closure systems with an efficacy of ≥96–99% in the case studies described above. Underlying these positive values is above all the ability of these adhesive fabrics to activate a haemostatic process by exploiting their chemical-physical characteristics. In particular, in the case of Glubran 2, the monomeric structure of cyanoacrylate, consisting of a double-carbon ethylene group with two electro-active reactive functions (cyano-CN and ester-COOR), reacting with the anions or radicals present in human blood or plasma (hydroxyl ions of H_2_O) through Michael addition reactions, would lead to the formation of a reactive carbanion or radical. The reaction continues until all accessible monomers are exhausted, or until the acid species stop the polymer growth. During polymerization, tissue amines can be added to the polymer chains as a Michael-type initiator, resulting in covalent bonds between the cyanoacrylates and the tissue. This would form a polymeric film that would adhere well to the body tissue, hold the wound edges together, and stop bleeding. This would result in a significant reduction of the local activated partial thromboplastin time (APTT), which is favourable with regard to the desired haemostasis as the shortening of the APTT leads to a haemostatic effect and contributes to glue-induced tissue adhesion. The results obtained in the operational and 3-day follow-ups of patients after treatment show no significant changes in prothrombin activity, fibrinogen, platelet count, or total and differential leucocyte count after treatment with Glubran 2. In vivo applications of Glubran 2 also revealed its excellent haemostatic and adhesive properties, in particular in binding biological tissues to each other or to prosthetic implants; the adhesion appeared tenacious immediately after application and consolidated its strength during the completion of the polymerisation process in various chemical-physical phases (liquid, gel-like, film), guaranteeing rapid and effective results in vascular surgery. The results obtained from the use of Arista^TM^AH showed that the porous hydrogel microspheres constituting the powder, once dispersed at the wound site, act as a molecular sieve, capable of contacting and activating platelets, leading to rapid haemostasis through the absorption of water and the concentration of coagulation factors. The powder’s powerful osmotic action dehydrates and turns blood into gel, accelerating the normal coagulation process. Evidence from applications of these sealants in the surgical environment has shown that both are safe and effective both as closure devices and as substances capable of stimulating the haemostatic process. Specifically in the surgical cases analysed, it was observed that Glubran 2 is a device that is particularly suitable for treating lesion sites where there is high blood pressure (pseudo aneurysm) or where the risk of infection is high (PICC stabilization), while Arista^TM^AH is a device that is particularly useful for sealing lesion sites with low blood pressure, such as FAVs, as it helps control venous, capillary, and arteriolar bleeding in general. The failures that occurred in the patient samples examined were more related to complications of the procedure than to the use of sealants. Specifically in the case of FAVs, the discrepancies in luminal diameters between artery and vein (the vein calibre considered ‘minimum’ for packing the fistula for haemodialysis is at least 3 mm) can cause the anastomosis on the distal radial artery to predispose to ischaemia of the hand (so-called artery steal). The risk is particularly high in elderly and diabetic patients (approx. 5% of our cases = 1 case) due to the parietal sclerosis of the arterial vessel being poorly compensated by the ulnar artery, which is also affected by atherosclerosis of the intima. One of the manoeuvres to optimize flow in a fistula is to avoid acute angles when the vessels are anastomosed, but when the radial artery is sclerosed acute angles cannot be consistently avoided. it is therefore preferred to perform a termino-lateral anastomosis. Another failure that may occur is venous thrombosis at the arterial branch of the fistula (0% of cases in our study). In the application of the PICC, catheter occlusion occurred on average 16 days after catheter placement in an average of 15% of patients (*n* = 4 patients). It is assumed that healthcare workers did not always follow the instructions for the prevention of catheter obstruction and perhaps did not pay attention to the warning signs requiring timely preventive measures, or the patient did not adequately comply with the recommended drug therapy (low molecular weight heparin) or did not carry out impermissible mechanical manoeuvres (physical exertion, lifting weight). Regarding infections, we found a PICC-related BSI rate of 0.42 per 1000 PICC days, which is lower than that reported in the literature. This could be related to the application of Glubran 2, which seals the catheter insertion site by isolating it from the external environment. With regard to the pseudoaneurysm, no arterial occlusion was ever found in the 10 patients treated (0% arterial occlusion) and wound dehiscence with edge infection and fibrin formation in 2% of the cases (two patients) was thought to be due to the excessive weakness of the skin, which was punctured then stretched by the pseudoaneurysm and then incised in the operating theatre and sutured.

No adverse reactions related to the occurrence of haemorrhage and/or allergic reactions have been reported in the literature from in vitro tests and in vivo application of Arista^TM^AH, as also demonstrated by the clinical cases we analysed. At discharge and 7 day follow-up, no patients reported systemic allergic reactions (no tearing or itchy eyes, no nasal discharge, no rhinorrhoea, no sneezing) or local allergic reactions (urticaria, itchy skin). With regard to the clinical cases we considered for the application of Glubran 2, it can be stated that no distal leakage of glue or thrombotic material occurred during or after the procedure. No haemorrhage, non-targeted glue embolization, stenosis or thrombosis, allergic reaction, infection, skin ischaemia, or rupture of the pseudoaneurysm (Glubran 2) occurred. All these were assessed with intraoperative echocolour doppler, postoperatively 1 day after the procedure and then at discharge after 7 days.

Numerous works are reported in the literature in which Glubran 2 and Arista are used as closure systems in different clinical applications. For example, in a work conducted by Grange et al., a 62-year-old male patient with a diagnosis of multi-system atrophy underwent a CT scan that revealed a pseudoaneurysm of the left upper epigastric artery. After unsuccessful surgical treatment, the patient was successfully treated with percutaneous embolisation using Glubran 2 under ultrasound control. The sealant proved to be safe and effective without inducing toxicity [28].

In a paper conducted by Athanasiou et al., a 9-year-old patient who came to hospital with vomiting and headache underwent CT and digital subtraction angiography, which revealed intracranial pseudoaneurysms in the frontoparietal branch of the right middle cerebral artery in segment M2. It was decided to proceed with urgent endovascular repair to occlude the pseudoaneurysm and the mother artery using Glubran 2. The postoperative course presented no major complications, and the patient was fully conscious and cooperative, with no focal neurological deficits [29].

Arista has also been used as a haemostatic sealant in several clinical applications. For example, a work conducted by Gleason et al. [30] evaluated the efficacy of Arista and its possible influence on rates of haematoma formation and wound infection in primary total knee arthroplasty. The results showed that this device has no clinically significant effect on the rates of haematoma formation or wound infection in this type of surgery and shows less than average bleeding control. In other works, however, it is reported that Arista, for example in cardiothoracic surgery, promotes a reduction in haemostasis time and less postoperative chest tube emission without an increase in complications [31], and has been proven to be safe and effective in living donor nephrectomies [32].

Antisdel et al. have conducted several studies in the otolaryngology field, where they reported less postoperative bleeding after sinus surgery and zero complications with regard to postoperative healing in the sinus cavity [33].

## 4. Materials and Methods

### 4.1. Study Design

One surgical glue and one powder were tested for internal use: Glubran 2 (supplied by General Enterprise Marketing GEM S.R.L, Viareggio, Lucca, Italy) and Arista^TM^AH (C. R. Bard, Inc.—Davol, Warwick, RI, USA). This study is an analysis of data collected retrospectively from patients of the Cosenza Hospital (Cosenza, Italy) who underwent open/endovascular surgical treatments requiring the use of Glubran 2; in a given sample of patients, Arista^TM^AH powder was applied at the site of anastomosis instead of glue, and a comparison of the two devices was carried out to test their stability and effectiveness at that site of action between January 2023 and September 2023. Specifically, these devices were effective in promoting surgical site closure by inducing haemostasis processes through different mechanisms.

Inclusion criteria were: (1) echo-guided venous access for endovascular placement of the catheter for drug infusion (sheath size 4Fr, MaxfloexpertVygon, Padova, Italy); (2) surgical access for arteriovenous fistula packing for haemodialysis; (3) surgical access for emptying and exclusion of iatrogenic pseudoaneurysm; (4) age over 18 years.

Exclusion criteria were: (1) age under 18 years; (2) pregnancy; (3) last haemodialysis session at least 36 h earlier; (4) bleeding risk higher than the recommendations of the CIRSE Standards of Practice on the management of peri-operative anticoagulation [34]; (5) presence of arterial thrombosis of the pseudoaneurysm and absence of echocolour and internal flowmetry; (6) history of superficial or deep vein thrombosis of the upper limb or ongoing thrombosis (PICC); (7) INR values > 1.5; platelet count < 70,000; (8) HB values < 8.0 g/dL; (9) white blood cell count > 12,000; (10) had Thromboplastine Partial Time values Activated (a-PTT) values > 40 s; (11) positive COVID-19 nasopharyngeal swab.

### 4.2. Data Collections and Procedure

All patients underwent pre-treatment evaluation with bedside ambulatory ultrasound using a SIEMENS S2000 ultrasound scanner (7.5 MHz linear probe) (Siemens Healthcare s.r.l. Milano, Italy), which was repeated peri-operatively in the operating room using a SONOSITE ultrasound scanner (7.5 MHz linear probe) (Fujifilm Healthcare, Milano, Italy), with transducer coverage under sterile conditions. In patients with iatrogenic arterial pseudoaneurysm, CT angiography (CTA) was performed. The choice of tissue adhesive depended on the operator’s preference according to the access technique used (endovascular and/or surgical), the site to be treated, and the type of material required to activate a haemostatic process at the site concerned. The surgical procedure for packing an arteriovenous fistula for haemodialysis was always performed in the operating theatre [35,36]. Having identified the site of anastomosis between the vein and the artery (radius cephalic/humeral basilic) by ultrasound, local anaesthetic was infiltrated with an incision of the skin at the level of the distal part of the left forearm. The cephalic vein was identified and isolated, which appeared to be of normal calibre (3–5 mm), and the radial artery was isolated (Figure 2).

After systemic heparinization by the anaesthesiologist with 2.000 I.U. of sodium heparin, proximal and distal clamping of the radial artery was performed, the radial artery was tomified, and the termino-lateral anastomosis was made with pro-lene 7.0. At declamping, blending bleeding from the anastomosis on the arterial side due to arterial calcifications was noted; therefore, Glubran 2 was applied to the anastomosis site at the arterial suture passage points (Figure 3A). In a sample of patients, Arista^TM^AH powder was applied at the site of anastomosis instead of glue, and a comparison of the two devices was carried out to test their stability and effectiveness at that site of action (Figure 3B).

The procedure for stabilization and fixation of the PICCs (peripherally inserted central catheters) took place in the angiography room after identification by ultrasound of the endovascular access site and exclusion of in situ thrombosis [37,38] (Figure 4). If the identified site was affected by venous thrombosis, the other arm was used.

In most cases, the catheter and venipuncture were performed in the middle third of the right arm, above the elbow crease, to identify the brachial or basilic vein. Under radioscopic guidance, the catheter was led up to the superior vena cava. When venipuncture on the forearm is not possible, it is necessary to use the veins on the back of the hand. No antibiotic therapy was administered as the ultrasound-guided procedure uses percutaneous access. The indication for the implantation of PICCs indicated whether patients must be given therapy for an infusion time of more than 6 days. The use of these devices has brought many benefits in addition to improving the quality of life of cancer patients, as it decreases pain and anxiety during the infusion of chemotherapy drugs, ensures greater safety in the administration of antiblastic drugs, and leads to a significant reduction in repeated venipuncture of patients and a consequent reduction in skin lesions and superficial phlebitis (chemical phlebitis). To stabilize the device and fix it at the percutaneous insertion point, Glubran 2 surgical glue was applied around the catheter (Figure 5a) and the PICC was stabilized (Figure 5b).

The procedure for the exclusion and emptying of the pseudoaneurysm took place in the operating theatre after ultrasound identification of the radial artery and the periarterial blood collection site (Figure 6).

After anaesthesia, an incision was made distal to the right forearm along the anatomical course of the artery [39,40]. Next, the radial artery upstream and downstream of the pseudoaneurysm was isolated, and the pseudoaneurysm was isolated from the surrounding tissues (Figure 7a) and incised to evacuate it (Figure 7b).

After emptying abundant clots, the iatrogenic lesions on the radial artery wall were in-divided and Glubran 2 haemostatic adhesive was applied (Figure 8).

### 4.3. Outcomes

The primary objective of this study was to evaluate the efficacy, safety, and benefits of using Glubran 2 and Arista^TM^AH in favouring the activation of haemostatic processes applied to the surgeon’s swift closure of arteriovenous fistulas and the repair of iatrogenic damage in patients undergoing surgical and endovascular treatment for arteriodosis, iatrogenic arterial pseudoaneurysm, arteriovenous fistula for haemodialysis, and stabilization and fixation of the PICC catheter. Primary technical success was defined by adequate haemostasis obtained in the peri-operative phase within a time frame of less than 5 min (in the case of Arista^TM^AH) and 2 min (in the case of Glubran 2) from the time of application of the product and temporary closure of the surgical site with minimal manual compression with sterile gauze. Secondary technical success was defined as haemostasis achieved after the above scheme or after a second subsequent period of manual compression with sterile gauze on the surgical site. Clinical success was defined by the absence of haematoma or bleeding-related peri-lesional collections at follow-up and the absence of peripheral neuropathy. Complications were classified according to SVS (Society for Vascular Surgery clinical practice guidelines) reporting standards and the bibliography can be found here [41].

Postoperative haematoma was defined as a palpable mass equal to or greater than 5 cm confirmed by ultrasound, failure to close arterial damage or anastomosis confirmed by echocolour doppler examination and supported by level II imaging (TC scan).

The diagnosis of neuropathy (radial nerve for radiocephalic fistula, median nerve for humerus basilic fistula, brachial nerve for PICC accesses) requires a specialized neurological consultation.

### 4.4. Haemostasis Evaluation

After removing the gauze, each lesion was qualitatively assessed for haemostasis at 2, 3, 4, 5, 6, 7, and 10 min after application using the validated bleeding scale. After the 10 min haemostasis assessment, the lesions were irrigated with saline and were visually assessed to remove excess material not incorporated into the blood clot (Arista^TM^AH) (Figure 9). In the case of Glubran 2, each lesion was qualitatively assessed for haemostasis at 2, 3, 4, and 5 min after application using the validated bleeding scale. The operator assessed the patency of the artery by appreciating the flow downstream of the site of use to avoid ‘STOP’ to blood flow should the haemostatic device migrate into the artery at the site of haemostasis, or should the anastomosis become excessively ‘stiff’ due to the use of the adhesive.

### 4.5. Statistical Analysis

Statistical analyses were conducted on all patients recruited for this study. The collected data were analysed using one-way ANOVA. In particular, the software SPSS for Analytics 25 (IBM, New York, NY, USA) was used for this analysis [42,43]. *p*-values < 0.05 and <0.001 were considered statistically significant for the aforementioned tests.

## 5. Conclusions

The results obtained show us that the two haemostatics Glubran 2 and Arista^TM^AH are ≥96–99% effective as closure systems in patients undergoing vascular surgery. In the cases we analysed, it was observed that Glubran 2 is a particularly suitable device for the treatment of lesion sites where high blood pressure is present (pseudo aneurysm), with a success rate of ≤97% or, where the risk of infection is high (PICC stabilisation), with almost 100% success.

Arista^TM^AH, on the other hand, proves to be a useful device for sealing lesion sites with low blood pressure, such as FAVs, showing a success rate of ≥94% as it helps to control bleeding at venous, capillary, and arteriolar levels. This means that the sealants promote haemostasis depending on the type of vascular lesion site and the blood flow conditions there, with success rates also relating to their chemical-physical characteristics (e.g., high porosity, high water absorption rate) and the absence of adverse reactions.

The only failures that have occurred in the patients under investigation are related to complications induced by the surgical procedures, errors on the part of healthcare professionals, and failure to continue the therapies prescribed to patients during discharge and/or follow-up.

For this reason, we can also consider these devices safe and effective in vivo applications.

## Data Availability

Data is contained within the article.

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
