# Peer review of "Synthetic Haemostatic Sealants: Effectiveness, Safety, and In Vivo Applications"

_pharmaceuticals, 2024, doi:10.3390/ph17030288_

Round 1

Reviewer 1 Report

Comments and Suggestions for Authors

see file

Comments on the Quality of English Language

There were some grammatical errors that made it difficult to interpret some of the information.

Author Response

Reviewer 1.

As stated in the Introduction:

“The aim of this work was to evaluate the effectiveness and safety of the two devices in the surgical field by differentiating their application based on the pathology treated and the hemostasis mechanism exerted on it.”

However, in the Conclusions it was stated:

The results of the present investigation show that the application of Glubran 2 and AristaTMAH as surgical wound closure systems is effective and safe in achieving hemostasis of the affected site in patients undergoing surgical treatment. The success achieved was≥94%on anastomoses of arteriovenous fistulas;100% on stabilization of Picc catheters; ≤97% on pseudoaneurysms. This allowed us to appreciate the versatility of the devices as sealants in vascular surgery.

Therefore, the conclusion does not really do what the intent of the paper is. It is also unclear why the psudoaneurysms (<97%) was considered a good result. The discussion, however, includes what is considered success (in many cases listed in Table 5) as well as the differences between the two geatments. It is also unclear what the statistics showed, since results were given in % success and the sample sizes might have been too low to be sensitive enough to pick up differences that would lead to success or failure.

Again, the discussion did a good job of summarizing the results. It gave a good description of the mechanisms as well as potential causes of failure of the procedures not directly caused by the adhesives. The conclusion, therefore, should have included some of the discussion related to:

“Differentiating their application based on the pathology treated”.

This essentially would be including what one was better for (high and low pressure was one thing mentioned) and what is needed to prevent failures for each.

The conclusion was modified and rewritten according to the reviewer's requests. The statistical analysis was also revised and reported ex novo in the text.

Reviewer 2 Report

Comments and Suggestions for Authors

The paper authored by Curcio and colleagues explores a study involving patients with various vascular diseases (FAV, pseudoaneurysm, and PICC application), employing two distinct haemostatic sealants. The objective is to evaluate the safety, advantages, and the efficacy of both sealants in activating the haemostatic process at the affected site, considering their chemical-physical characteristics.

The manuscript exhibits commendable organization and holds promise for significant contributions to its field. The results and conclusions are well-supported by in-vivo experiments involving patients who underwent open/endovascular surgical treatments, along with a meticulous evaluation of hemostasis using the two sealants.

In my opinion, the manuscript is suitable for publication in its current form.

Author Response

We thank the reviewer for positively evaluating our work

Reviewer 3 Report

Comments and Suggestions for Authors

This is a clinical efficacy and safety study of a finished drug. This requires further improvement in this study, as the sample size is clearly insufficient as a clinical study. If used as an evaluation of the efficacy and safety of a new drug, a detailed list of the drug's ingredient ratios, molecular structure characterization, and animal experimental results should be provided, rather than clinical data. Therefore, I believe that this part of the research is currently not suitable for publication and requires more clinical sample data.

Author Response

We have made some changes to the manuscript. However, we thank the reviewer for the time dedicated to reviewing the manuscript
